# STYLE UNLEARNING IN DIFFUSION MODELS

## ABSTRACT

For diffusion models, machine unlearning is crucial for mitigating the intellectual property and ethical challenges arising from unauthorized style replication. However, most existing unlearning methods struggle to completely remove styles while preserving generation quality, as their erasure mechanisms rely on the noise distribution where style and content are intrinsically entangled. To address it, we propose **S**tyle **U**nlearning in **D**iffusion **M**odels (SUDM), a novel framework based on hybrid-attention distillation, where cross-attention provides style-agnostic supervision to self-attention for targeted style erasure. By leveraging the structural distinctions within attention component, SUDM enables more effective destylized modeling compared to previous work. To further ensure content preservation and robust generalization, we introduce query consistency and parameter consistency losses into the overall objective function. Finally, extensive experiments and user studies on Stable Diffusion demonstrate that SUDM achieves more thorough style erasure with minimal quality degradation, outperforming existing unlearning methods in both visual fidelity and precision. Our code is available in the supplementary materials.

## 1 INTRODUCTION

Text-to-image diffusion models Ho et al. (2020); Dhariwal & Nichol (2021b) are trained on LAION-5B Schuhmann et al. (2022), a massive but minimally curated collection of image-text pairs Carlini et al. (2023). As a result, they can generate photorealistic images Yu et al. (2022); Gafni et al. (2022); Chang et al. (2023); Xu et al. (2023b) and imitate artistic styles, especially when guided by prompts like "art in the style of [artist]" Gafni et al. (2022); Saharia et al. (2022); Yu et al. (2022); Somepalli et al. (2023); Chang et al. (2023). Nevertheless, alongside their impressive abilities, these models can produce unauthorized or harmful content—including copyrighted art Somepalli et al. (2023); Shan et al. (2023), explicit imagery Schramowski et al. (2023b), and deepfakes Carlini et al. (2023)—which raises serious legal and ethical concerns. As illustrated in Fig. 1, Stable Diffusion CompVis (2022) can fabricate replicas of Van Gogh's distinctive style, posing significant threats to the authenticity and integrity of artistic works. Lawsuits from artists such as Kelly McKernan against Stable Diffusion underscore these challenges Setty (2023). Accordingly, it is imperative to unlearn specific styles or visual modes embedded in pre-trained diffusion models, thereby upholding established ethical norms in AI development.

Existing methods mitigate the unauthorized replication of artistic styles in diffusion models fall into two groups: *1) Training-free methods* which perform real time erasure during inference without fine-tuning the pre-trained diffusion model CompVis (2022); Schramowski et al. (2023a); Li et al. (2023). Although these methods computationally efficient, their erasure performance is limited, primarily because they struggle to eliminate style information that is intrinsically embedded in the model configuration. *2) Training-based methods*, such as ESD Gandikota et al. (2023), SPM Lyu et al. (2023),ConAbl Kumari et al. (2023a), and UCE Gandikota et al. (2024), mainly focus on erasing semantic concepts by fine-tuning pre-trained diffusion models. However, since they rely on the predicted noise distribution—which is insensitive to abstract and intricate style patterns—their ability to erase specific stylistic representations remains limited, often accompanied by degraded generation quality. Therefore, the development of unlearning techniques tailored explicitly for stylistic representations in diffusion models remains an urgent and open challenge, with no existing solutions to date.

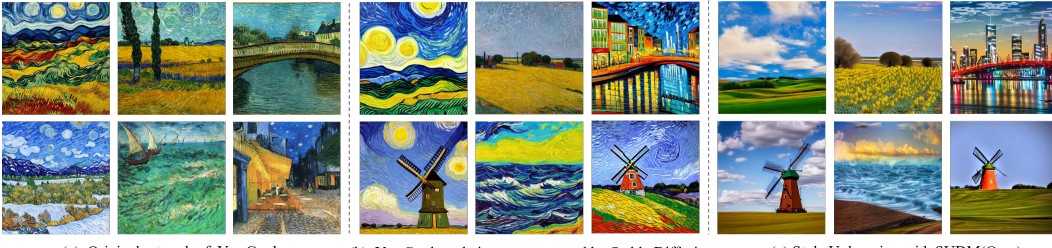

(a) Original artwork of Van Gogh   (b) Van Gogh-style images generated by Stable Diffusion   (c) Style Unlearning with SUDM(Ours)

Figure 1: (a): Original Van Gogh artworks. (b): Images generated by Stable Diffusion models that mimic Van Gogh's style. (c) Results after applying SUDM to remove Van Gogh's style from (b). This comparison illustrates the model's capability to reproduce the visual characteristics of a specific artist as well as the effectiveness of SUDM in style unlearning.

To address this problem, we propose a novel framework, **S**tyle **U**nlearning in **D**iffusion **M**odels (SUDM), by leveraging **Hybrid-Attention Distillation(HAD)** module with joint attention mechanisms and distillation mechanisms. The rationale behind this approach stems from a key trait on attention mechanisms validated by Hertz et al. (2024); Jeong et al. (2024); Zhou et al. (2025) that the key (K) and value (V) components of self-attention encode style-related representations whereas the query (Q) component predominantly preserves semantic content. Accordingly, it is feasible by leveraging self-attention to capture the model's representations on the specific styles. Simultaneously, it is workable by employing cross-attention to achieve style-neutral representations with the consistent content semantics. Thereafter, we design a distillation loss to align these two attention outputs, which facilitates the model to discard style-specific information without distorting content semantics. Specifically, given a style-neutral reference image and a stylized prompt such as "The night by Van Gogh", hybrid-attention distillation leverages the key ($\mathbf{K}$) and value ($\mathbf{V}$) from the reference image as style-neutral anchors, and the query ($\mathbf{Q}$) from the generated image to encode content information. Then, a cross-attention mechanism between $\mathbf{Q}$ and the style-neutral $\mathbf{K}$, $\mathbf{V}$ guides the self-attention in the generated image to unlearn the style.

To further preserve semantics, we enforce query consistency between the reference and stylized generations, and ensure generalization through parameter consistency during training. Consequently, as shown in Fig. 1(c), SUDM effectively erases style information with minimal impact on semantic content and generation performance.

Our main contributions are summarized as follows:

- We propose a novel framework for style unlearning in diffusion models by leveraging hybrid attention distillation module (HAD), query consistency, and parameter consistency techniques, named SUDM. To the best of our knowledge, it is the first unlearning technique tailored for style removal from diffusion models.

- Innovatively, our HAD module aligns stylized and style-neutral representations, to facilitate targeted style removal while preserving semantic content. Unlike the alignment in noise space, the alignment in representation space is capable to capture non-interfering embedded style and content, thereby benefiting for more precise removal of style.

- Extensive experiments depict that our method can effectively unlearn the style while preserving the generation performance of other concepts.

## 2 RELATED WORK

Machine unlearning Xu et al. (2023a) has recently gained increasing attention in diffusion models to mitigate the retention of undesirable knowledge, such as copyrighted content Somepalli et al. (2023), biased representations Jiang et al. (2024), or specific artistic styles Shan et al. (2023). Existing approaches are broadly categorized into *training-free* Schramowski et al. (2023a); Yang et al. (2024); Wang et al. (2025) and *training-based* methods Gandikota et al. (2023); Kumari et al. (2023a); Lyu et al. (2023); Gandikota et al. (2024). Training-free methods aim to unlearn or avoid generating undesired concepts without fine-tuning the pretrained diffusion model, and they broadly fall into

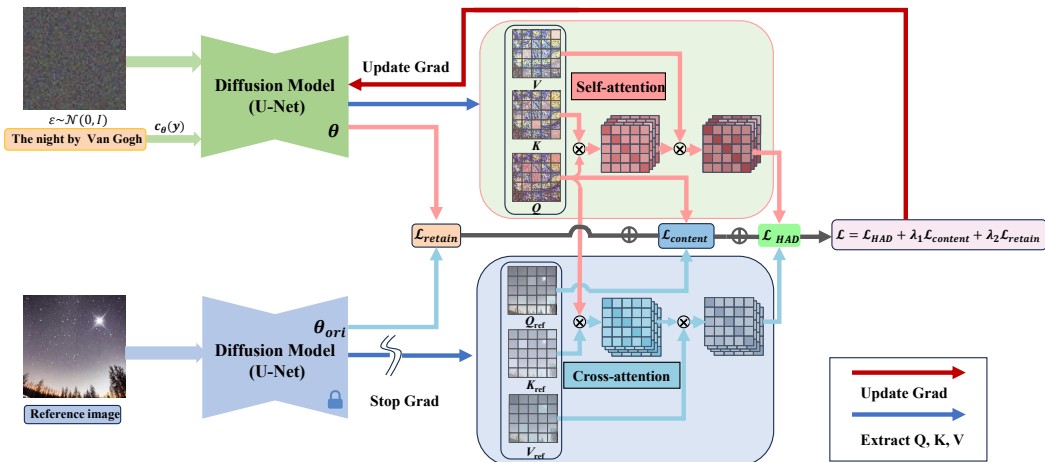

Figure 2: Overview of the proposed SUDM framework. Given a stylized prompt and a style-neutral reference image with shared content, SUDM extracts the self-attention response and applies distillation to remove style-specific representations while preserving semantic content.

inference-guided and pruning-based approaches. *Inference-guided methods* guide the generation process at inference time to reduce the influence of specific concepts. For instance, NP CompVis (2022) and SLD Schramowski et al. (2023a) modify the classifier-free guidance mechanism to diminish the effect of undesired content. A more recent inference-guided method, AdaVD Wang et al. (2025), leverages classical linear algebraic orthogonal complement operations implemented in the value space of cross-attention layers , enabling precise disentanglement of target semantics from non-target ones. However, due to the entanglement of style and content features in the model's latent space, these methods often result in limited precision and reduced image quality. *Pruning-based methods* identify and deactivate internal components associated with undesired concepts. For example, ConceptPrune Chavhan et al. (2024) applies activation-based importance scoring or learns binary masks to turn off neurons or attention heads related to certain concepts. While pruning maintains model size and is computationally efficient, it often lacks semantic precision and struggles to generalize beyond the specific examples used. Additionally, these methods fall short in handling abstract or widely distributed representations, such as artistic styles.

Training-based methods mainly focus on fine-tuning the model to unlearn target concepts. For example, ESD Gandikota et al. (2023) aligns the output distribution of target concept with that of an empty prompt, while FMN Zhang et al. (2024) progressively erases the concept during inference by re-steering attention maps within the U-Net's cross-attention layers. However, these approaches lack explicit mechanisms to preserve the non-target concepts, often causing side effects such as training instability, semantic drift, and degraded generation quality for unrelated content. To mitigate this, ConAbl Kumari et al. (2023a) aligns the predicted noise between target and anchor concepts and utilizes a regularization loss to maintain anchor integrity. UCE Gandikota et al. (2024) edits cross-attention projections tied to the target text embeddings. Similarly, SPM Lyu et al. (2023) proposes a latent anchoring strategy combined with a similarity-aware retention loss to better preserve the surrogate concepts. While these methods enhance the stability of concept unlearning and reduce unintended degradation of unrelated concepts, they face challenges in style unlearning due to the diffuse, spatially variable, and tightly entangled nature of styles with semantic content. Therefore, we propose a style unlearning method for diffusion models that leverages cross-attention to guide self-attention with style-agnostic supervision for targeted style erasure.

## 3 METHOD

This section introduces SUDM which comprises three complementary components: hybrid-attention distillation for removing style-specific representations, content preservation for retaining semantic content, and generalization preservation for preventing performance degradation on generations of

unrelated concepts. An overview of SUDM is illustrated in Fig. 2. We begin by introducing the problem setup, and then detail the design of each component.

## 3.1 PRELIMINARY

**Diffusion Models** Ho et al. (2020); Song et al. (2021) learn to reverse a Markov process that gradually adds Gaussian noise to an image $x_0$, producing $x_t$ at each timestep $t$. This forward process is defined as $x_t = \sqrt{\alpha_t}x_0 + \sqrt{1-\alpha_t}\epsilon$, where $\alpha_t$ is a fixed or learned variance schedule and $x_T \sim \mathcal{N}(0, I)$. A denoising network $\epsilon_\theta(\cdot)$ is trained to recover $x_{t-1}$ from $x_t$, optionally conditioned on additional inputs $c$ (e.g., text). Image generation is performed by iterative denoising from $t = T$ to 0. The training objective is to predict the added noise $\epsilon$:

$$\mathcal{L}(x, c) = \mathbb{E}_{\epsilon \sim \mathcal{N}(0,1), x, c, t} \left[ \|\epsilon - \epsilon_\theta(x_t, c, t)\|_2^2 \right]. \tag{1}$$

**The attention mechanism in diffusion models** plays a crucial role in capturing long-range dependencies and semantic correspondences during image generation Vaswani et al. (2017). Given an input feature map $X$, the attention module first computes three learnable linear projections: query ($Q$), key ($K$), and value ($V$), formulated as:

$$Q = XW^Q, \quad K = XW^K, \quad V = XW^V, \tag{2}$$

where $W^Q$, $W^K$, and $W^V$ are trainable projection matrices. For brevity, we abbreviate Attention($\cdot$) as Attn($\cdot$) in the following. These projections are used to compute pairwise affinities between elements in the feature space. The attention output is then obtained by:

$$\text{Attention}(Q, K, V) = \text{softmax}\left(\frac{QK^\top}{\sqrt{d}}\right)V, \tag{3}$$

where $d$ denotes the dimensionality of the queries and keys, and the softmax operation ensures that attention weights are normalized. This mechanism allows the model to adaptively aggregate contextual information across spatial locations, which is particularly beneficial in diffusion models for generating semantically coherent and structurally consistent outputs Dhariwal & Nichol (2021a).

## 3.2 PROBLEM SETUP

Let $\epsilon_\theta(\cdot)$ be a pre-trained latent diffusion model (e.g., Stable Diffusion CompVis (2022)) capable of generating images from text prompts. Given a stylized prompt $c$, our objective is to remove $\epsilon_\theta$'s ability to generate images in the style associated with $c$ (e.g., Van Gogh), such that images conditioned on style-indicative prompts no longer exhibit the corresponding distinctive characteristics, while still preserving content fidelity and generalization to other styles. To facilitate style unlearning, we construct a training dataset $\{(c, I_{\text{ref}})\}$, where $I_{\text{ref}}$ is a reference image that conveys the same semantic content as $c$ but in a neutral style, distinct from the one implied by $c$ (e.g., photorealism or Pixar-style).

## 3.3 HYBRID-ATTENTION DISTILLATION

In latent diffusion models such as Stable Diffusion, Cross-attention and Self-attention play distinct yet complementary rolesLiu et al. (2024). Cross-attention injects textual semantics into the image latent space, aligning generation with the prompt, whereas self-attention operates solely on visual latents to capture long-range spatial dependencies and regulate stylistic consistency across the image. Building on this observation, recent works Hertz et al. (2024); Jeong et al. (2024); Chung et al. (2024) in image editing have exploited self-attention for style manipulation where the attention is computed using the content query $Q_c$ and the style key $K_s$ and value $V_s$, hence term *KV injection*. Their hypothesis is that $Q$ encapsulates the content within an image, while $K$ and $V$ represent the style information. This assumption is supported by the experiments in image editing research, which demonstrate that semantic structures are mainly preserved in queries, while stylistic attributes are embedded in keys and values.Huang et al. (2025) Based on these insights, we propose a hybrid-attention distillation module to manipulate the self-attention mechanism for targeted style erasure.

Specifically, we extract the query, key, and value representations from the image generated using a stylized prompt $P$, denoted as $Q_l^t, K_l^t, V_l^t$, at layer $l$ and timestep $t$. Similarly, we obtain

$Q_l^{\text{ref},t}, K_l^{\text{ref},t}, V_l^{\text{ref},t}$ from a style-neutral reference image. Among these, $K_l^{\text{ref},t}$ and $V_l^{\text{ref},t}$ serve as the anchors for style removal. We then define the hybrid-attention distillation loss at each selected layer and timestep as:

$$\mathcal{L}_{\text{HAD}} = \left\| \text{Attn}(Q_l^t, K_l^t, V_l^t) - \text{Attn}(Q_l^t, K_l^{\text{ref},t}, V_l^{\text{ref},t}) \right\|_2^2. \tag{4}$$

The first term in (4) represents the original self-attention of the image generated from the stylized prompt, while the second term denotes a cross-attention operation that replaces the key and value matrices with those from the style-neutral reference image to unlearn stylistic bias. This design ensures that the model focuses on erasing style-related features, while preserving the semantic content encoded primarily in the query representations.

## 3.4 Content Preservation

While the proposed $\mathcal{L}_{\text{HAD}}$ effectively removes style-specific features, it may inadvertently compromise the semantic content of the image. To mitigate this issue, we introduce a content preservation mechanism by enforcing consistency between the query representations of the generated image (from the stylized prompt) and those of the reference image at each timestep $t$ and layer $l$, as formalized in (5).

$$\mathcal{L}_{\text{content}} = \left\| Q_l^t - Q_l^{\text{ref},t} \right\|_2^2. \tag{5}$$

As the query representations predominantly encode semantic content Zhou et al. (2025), enforcing alignment between the query matrices of the stylized and reference images facilitates semantic preservation and alleviates potential compromise caused by style erasure.

## 3.5 Generalization Preservation

Inevitably, the joint optimization of hybrid-attention distillation and content preservation objectives induces a deviation of model parameters from those of the original pre-trained model. Such a shift may adversely affect the model's ability to generalize to other styles or content. To address this issue and retain generalization during effective erasure, we introduce a retain loss that regularizes the parameters toward the original pre-trained weights:

$$\mathcal{L}_{\text{retain}} = \left\| \theta - \theta_{\text{ori}} \right\|_1, \tag{6}$$

where $\theta_{ori}$ denote the original parameters of the model. This regularization encourages the model to preserve its generalization ability while unlearning the target style.

Then, the total loss is defined as:

$$\mathcal{L}_{\text{total}} = \mathcal{L}_{\text{HAD}} + \lambda_1 \mathcal{L}_{\text{content}} + \lambda_2 \mathcal{L}_{\text{retain}}, \tag{7}$$

where $\lambda_1$ and $\lambda_2$ are hyperparameters that balance the contributions of different objectives.

This formulation (7) unifies three complementary components: *1)* style unlearning through hybrid-attention distillation, which selectively suppresses style-related information; *2)* content preservation via alignment of query representations, which encourages the model to retain content consistency with the reference image; and *3)* parameter regularization to maintain model generalization, mitigating potential overfitting to the erased style. Together, these components enable effective and controllable style erasure while maintaining high fidelity to the original content and preserving the generative capacity of the model across diverse inputs.

During training, we apply the above three losses across multiple timesteps and selected attention layers of the diffusion model. The hybrid attention and query alignment are computed per layer. This design allows targeted style forgetting while retaining the model's generation ability on unrelated concepts. We summarize the training process in Alg 1.

## 4 Experiments

### 4.1 Experiment Settings

**Datasets.** We employ four datasets, each corresponding to a distinct artistic style: Van Gogh, Claude Monet, Pablo Picasso, and Rembrandt. Each training set contains 50 stylized prompts that

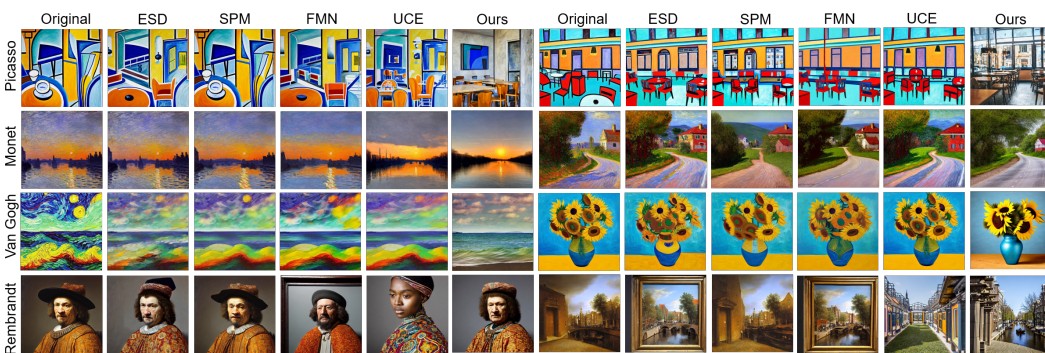

Figure 3: Visualization of unlearning on Van Gogh, Monet, Picasso, and Rembrandt shows our method removes target styles while preserving content, outperforming baselines.

| Methods | Monet | | Picasso | | Van Gogh | | Rembrandt | |
|---|---|---|---|---|---|---|---|---|
| | CS ↓ | FID ↑ | CS ↑ | FID ↓ | CS ↑ | FID ↓ | CS ↑ | FID ↓ |
| ESD | 66.12 | 90.30 | 66.55 | 53.95 | 68.17 | 91.58 | 67.23 | 76.91 |
| FMN | 57.78 | 113.29 | 66.52 | 48.58 | 65.82 | 111.34 | 67.22 | 78.48 |
| UCE | 63.27 | 93.68 | **69.15** | 38.17 | 72.88 | **44.69** | 69.43 | 48.17 |
| SPM | 71.64 | 57.58 | 67.58 | **27.46** | 71.95 | 50.39 | 68.98 | **40.62** |
| Ours | **57.37** | **120.29** | 67.75 | 72.56 | **72.95** | 58.69 | **69.86** | 93.26 |

Table 1: Results of unlearning the Monet style while preserving the styles of Picasso, Van Gogh, and Rembrandt. Lower CS and higher FID indicate better erasure; higher CS and lower FID indicate better preservation. Best and second-best results are in **bold** and underlined, respectively.

capture the visual characteristics of the target artist while covering a diverse range of semantic scenes (e.g., landscapes, portraits, objects). The test set includes 20 prompts that are disjoint from those used in training.

To construct the reference images in our training set, we start by extracting a base prompt from each stylized prompt (e.g., from the starry night by Van Gogh, we derive the night with stars by removing style-specific keywords). We then pair the base prompt with a negative prompt that specifies the artist (e.g., negative prompt = "Van Gogh") to generate a reference image that retains the high-level content but excludes the targeted artistic style. The resulting reference images, together with their original stylized prompts, serve as the training data in our study.

**Baselines.** We compare our method with four latest approaches including ESD Gandikota et al. (2023), FMN Zhang et al. (2024), UCE Gandikota et al. (2024), and SPM Lyu et al. (2023). The Implementation details of SUDM are provided in A.2

**Evaluations.** We adopt a leave-one-style-out evaluation, where each of the four datasets (Van Gogh, Monet, Picasso, Rembrandt) is held out in turn as the target style to forget, with the others retained. This rotational setup enables a thorough examination of the method's ability to selectively forget one style while preserving the others.

To assess unlearning and preservation performance, we employ two standard metrics: *CLIP Score (CS)* Hessel et al. (2021); Beaumont (2022); Kumari et al. (2023b) and *Fréchet Inception Distance (FID)* Heusel et al. (2017). CS quantifies the similarity between generated images and the target prompt, while FID measures the distributional distance between images generated post-unlearning and those produced by the original model. For the forgotten style, lower CS and higher FID values indicate more effective unlearning. In contrast, for the preserved styles, higher CS and lower FID values suggest better retention of the model's generative capabilities. For each evaluation, we generate 400 images using 20 test prompts, and report the average performance across the datasets.

| Methods | Van Gogh | | Picasso | | Monet | | Rembrandt | |
|---|---|---|---|---|---|---|---|---|
| | CS ↓ | FID ↑ | CS ↑ | FID ↓ | CS ↑ | FID ↓ | CS ↑ | FID ↓ |
| ESD | 65.94 | 116.97 | **67.01** | 53.95 | 74.45 | 53.71 | 68.23 | 73.71 |
| FMN | 63.48 | 124.16 | 65.82 | 54.04 | 71.43 | 59.95 | 67.73 | 70.51 |
| UCE | 61.90 | 120.89 | 67.04 | **33.59** | 74.43 | **27.53** | 68.95 | **51.03** |
| SPM | 63.91 | 111.18 | 66.27 | 39.67 | 73.44 | 43.81 | 67.99 | 61.64 |
| Ours | **61.07** | **163.31** | 66.87 | 67.69 | **75.18** | 53.53 | **70.15** | 93.26 |

Table 2: Results of unlearning the Van Gogh style while preserving the styles of Picasso, Monet, and Rembrandt. Lower CS and higher FID indicate better erasure; higher CS and lower FID indicate better preservation. Best and second-best results are in **bold** and underlined, respectively.

| Methods | Picasso | | Van Gogh | | Monet | | Rembrandt | |
|---|---|---|---|---|---|---|---|---|
| | CS ↓ | FID ↑ | CS ↑ | FID ↓ | CS ↑ | FID ↓ | CS ↑ | FID ↓ |
| ESD | 62.91 | 94.44 | 71.18 | 81.23 | 73.00 | 49.45 | 69.00 | 71.29 |
| FMN | **61.08** | 103.58 | 66.64 | 104.06 | 73.20 | 111.34 | 68.15 | 70.42 |
| UCE | 61.45 | 97.38 | 72.54 | 53.46 | **79.41** | 35.86 | 69.85 | 59.23 |
| SPM | 62.84 | 74.21 | 72.44 | **48.09** | 74.73 | **20.86** | 68.95 | **45.10** |
| Ours | 61.87 | **151.11** | **72.56** | 126.65 | 74.33 | 84.43 | **72.12** | 118.91 |

Table 3: Results of unlearning the Picasso style while preserving the styles of Monet, Van Gogh, and Rembrandt. Lower CS and higher FID indicate better erasure; higher CS and lower FID indicate better preservation. Best and second-best results are in **bold** and underlined, respectively.

**Single-style unlearning.** We assess SUDM's performance in unlearning a target style while retaining quality for others, with results for Monet, Van Gogh, Picasso, and Rembrandt presented in Tables 1–4. The visual effects of different unlearning methods are compared in Fig. 3.

The results reveal three main findings: *1)* As shown in Tables 1–4, SUDM consistently achieves the lowest CS when unlearning Monet, Van Gogh, and Rembrandt, and the highest FID when unlearning Monet, Van Gogh, and Picasso, demonstrating superior unlearning performance. From the visual comparisons in Fig. 3, SUDM also shows more thorough erasure of the target style, while others retain stylistic artifacts. For instance, ESD and SPM fail to remove Picasso's abstract textures, and FMN and UCE only achieve partial removal. *2)* SUDM also demonstrates strong generalization preservation. Specifically, it achieves the highest CS on Van Gogh and Rembrandt, and the second-highest CS on Picasso when Monet is the target of unlearning, indicating that it preserves non-target styles effectively during the unlearning process. *3)* SUDM exhibits a higher FID than most other methods in preserving non-target styles, primarily due to its emphasis on effectively erasing the target style, which inevitably compromises preservation fidelity. In contrast, the lower FID scores of other methods often result from insufficient erasure rather than superior generalization. As shown in the sensitivity analysis (Section 4.3), increasing $\lambda_2$ (preservation) and decreasing $\lambda_1$ (erasure) lowers the FID, highlighting the trade-off between forgetting and generalization. This trade-off reflects the strength of SUDM: it achieves effective erasure while maintaining competitive generalization on non-target styles.

**Multiply-style unlearning.** To further evaluate the scalability of SUDM, we consider a more challenging setting that requires simultaneously unlearning both Van Gogh and Monet styles. Notably, ESD is not applicable in this scenario due to its inability to handle multi-style unlearning, and thus is not reported. As shown in Table 5, SUDM significantly reduces the CS for both target styles while maintaining performance comparable to UCE, indicating effective removal of the stylistic features. These results demonstrate that SUDM not only supports multi-style unlearning but also achieves robust generalization.

## 4.2 ABLATION STUDY

We conduct ablation studies to assess each loss's contribution in SUDM for unlearning Van Gogh style while preserving Monet. Table 6 reports CS and FID scores, and Fig. 4 visualizes the ef-

| Methods | Rembrandt | | Van Gogh | | Monet | | Picasso | |
|---|---|---|---|---|---|---|---|---|
| | CS ↓ | FID ↑ | CS ↑ | FID ↓ | CS ↑ | FID ↓ | CS ↑ | FID ↓ |
| ESD | 63.81 | 102.12 | 71.46 | 69.20 | 72.75 | 54.50 | **68.74** | 47.83 |
| FMN | 64.48 | 121.54 | 68.26 | 85.93 | 72.52 | 54.99 | 66.65 | 48.98 |
| UCE | 65.18 | **171.59** | 73.09 | 58.18 | **74.90** | **46.60** | 68.10 | 43.42 |
| SPM | 68.99 | 55.73 | **73.24** | **47.83** | 71.06 | 62.91 | 68.49 | **36.87** |
| Ours | **63.39** | 132.15 | 71.98 | 108.48 | 73.05 | 60.74 | 68.16 | 78.25 |

Table 4: Results of unlearning the Rembrandt style while preserving the styles of Monet, Van Gogh, and Picasso. Lower CS and higher FID indicate better erasure; higher CS and lower FID indicate better preservation. Best and second-best results are in **bold** and underlined, respectively.

| Methods | Van Gogh | | Monet | | Picasso | | Rembrandt | |
|---|---|---|---|---|---|---|---|---|
| | CS ↓ | FID ↑ | CS ↓ | FID ↑ | CS ↑ | FID ↓ | CS ↑ | FID ↓ |
| ESD | - | - | - | - | - | - | - | - |
| FMN | 63.80 | 120.12 | 68.12 | 80.42 | 66.41 | 52.02 | 67.25 | 80.13 |
| UCE | **59.11** | **134.62** | **61.16** | **105.46** | 68.85 | 45.33 | 68.10 | 62.52 |
| SPM | 63.78 | 104.01 | 72.06 | 55.75 | **68.96** | **38.09** | 67.25 | **56.36** |
| Ours | 62.65 | 129.40 | 63.67 | 91.19 | 67.08 | 73.83 | **68.51** | 97.05 |

Table 5: Results of unlearning Van Gogh and Monet while preserving the styles of Picasso and Rembrandt. Lower CS and higher FID indicate better erasure; higher CS and lower FID indicate better preservation. Best and second-best results are in **bold** and underlined, respectively. ESD does not support multi-style unlearning ("-").

fects of removing $\mathcal{L}_{\text{HAD}}$, $\mathcal{L}_{\text{content}}$, and $\mathcal{L}_{\text{retain}}$. We further examine the impact of layer selection by progressively including self-attention layers from shallow to deep, as shown in Fig. 7,

**Effect of HAD loss $\mathcal{L}_{\text{HAD}}$.** Removing $\mathcal{L}_{\text{HAD}}$ notably weakens unlearning, as reflected by the higher CS to Van Gogh ($0.61 \rightarrow 0.72$) in Table 6. The results in Fig. 4 further confirm this: unlike SUDM (second column), the model without $\mathcal{L}_{\text{HAD}}$ (third column) fails to remove Van Gogh style.

**Effect of the content-preserving loss $\mathcal{L}_{\text{content}}$.** Ablating the content-preserving loss $\mathcal{L}_{\text{content}}$ leads to noticeable content corruption. As shown in Fig. 4, the original image contains a distinct sun element, which fails to appear when $\mathcal{L}_{\text{content}}$ is removed (see the fourth column). This results in both visual content distortion and semantic shift. The decline in CS for Van Gogh, as shown in Table 6, further underscores the importance of content preservation during unlearning.

**Effect of the retain loss $\mathcal{L}_{\text{retain}}$.** We evaluate $\mathcal{L}_{\text{retain}}$, which designed to limits parameter drift and preserve generalization on non-target styles. Removing it lowers CS for styles like Monet (Table 6) and fails to preserve Monet's style visually (Fig. 4, fifth column), highlighting its key role in maintaining generalization during unlearning.

**Effect of the selection of the self-attention layers.** We examine the effect of selecting different ranges of self-attention layers for optimization, as shown in Fig.7. The results reveal that the erasure improves with deeper inclusion, but full-layer optimization causes overfitting and quality degradation. Consequently, we choose to adopt the first ten layers as a balanced configuration.

### 4.3 SENSITIVITY ANALYSIS

To evaluate the sensitivity of the loss trade-off, we analyze the effects of $\lambda_1$ (content preservation loss $\mathcal{L}_{\text{content}}$) and $\lambda_2$ (generalization preservation loss $\mathcal{L}_{\text{retain}}$). Fig. 6 shows the CS and FID under varying $\lambda_1$ and $\lambda_2$ on the Van Gogh unlearning task, with the Monet style used to assess generalization to non-target styles.

| Variant | Van Gogh (Unlearning Target) | | Monet (Preservation Target) | |
|---|---|---|---|---|
| | CLIP ↓ | FID ↑ | CLIP ↑ | FID ↓ |
| w/o HAD | 72.65 | 136.07 | 73.36 | 88.95 |
| w/o $\mathcal{L}_{content}$ | 62.07 | 112.99 | 70.85 | 61.01 |
| w/o $\mathcal{L}_{retain}$ | 60.11 | 205.26 | 60.02 | 111.43 |
| SUDM (Ours) | 61.46 | 184.54 | 71.85 | 76.99 |

Table 6: Ablation results for unlearning Van Gogh while preserving Monet. For evaluation metrics: ↓ indicates lower values are better (better style unlearning for Van Gogh, better style preservation consistency for Monet), ↑ indicates higher values are better (stronger style erasure for Van Gogh, higher style fidelity for Monet).

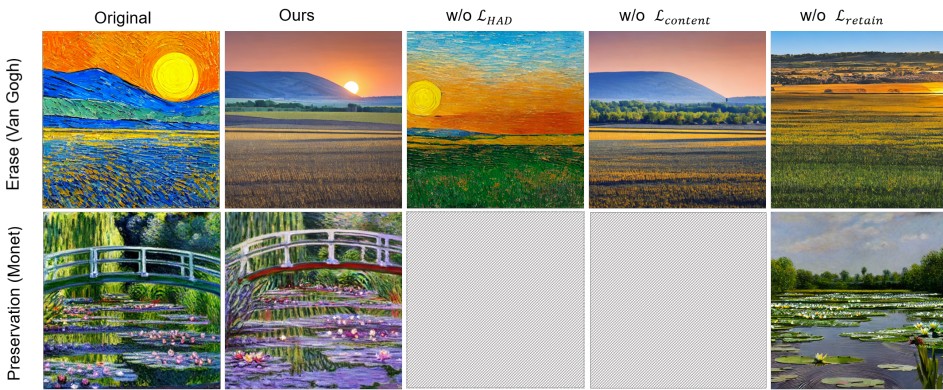

Figure 4: Visual illustration of component effects in SUDM. The top row shows Van Gogh style erasure, and the bottom row shows Monet style preservation. From left to right: (1) Original artwork. (2) full model (Van Gogh erased, Monet retained); (3) w/o $\mathcal{L}_{HAD}$ (Van Gogh not fully erased); (4) w/o $\mathcal{L}_{content}$ (content distorted, e.g., sun missing); (5) w/o $\mathcal{L}_{retain}$ (Monet style not preserved).

From Fig. 6, we have the following two key findings. *1)* As $\lambda_1$ increases (top subfigures), CS slightly decreases while FID rises sharply, indicating that stronger query consistency enhances target-style erasure(as indicated by a lower CS and higher FID on Van Gogh), but compromises generalization to non-target styles(as reflected by decreased CS and increased FID on Monet). *2)* As $\lambda_2$ increases (bottom subfigures), CS increases and FID decreases, suggesting that greater parameter consistency weakens erasure but improves generalization preservation.

Therefore, balancing erasure and generalization is crucial. To emphasize erasure, $\lambda_1$ can be increased but kept below 1 to avoid harming generalization. To preserve generalization, $\lambda_2$ should remain under $5 \times 10^{-4}$, as larger values result in a CS around 0.70, indicating incomplete erasure.

## 5 CONCLUSION AND LIMITATIONS

In this paper, we present **SUDM**, a novel framework for style unlearning in diffusion models. Unlike prior methods that leverage the predicted noise distribution and struggle to capture abstract stylistic features, **SUDM** exploits attention-based representations to achieve more precise style modeling. This enables selective removal of style-related representations while maintaining the semantic content. Extensive experiments on Stable Diffusion demonstrate that our method effectively erases specific artistic styles while minimizing degradation to unrelated concepts.

Despite its promising results, **SUDM** has several limitations. It assumes a clear separation between style and content in self-attention representations, which may not hold for abstract or entangled styles. Moreover, it depends on well-curated reference images, which may limit its applicability in real-world settings. Future work may investigate theoretical guarantees for the trade-off between style unlearning and generalization.

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

## A APPENDIX

### A.1 ALGORITHM SUDM

We present the algorithm SUDM as follows:

---

**Algorithm 1** **S**tyle **U**nlearning in **D**iffusion **M**odels

---

**Require:** Target stylized prompts $c$, reference images $I_{\text{ref}}$, pretrained model $\epsilon_{\theta_{ori}}$, model to train $\epsilon_\theta$, number of epochs $N$, $\epsilon \sim \mathcal{N}(0, I)$, set of layers $L_{\text{sel}}$, sampling steps $T_{\max}$.
1: **for** epoch $i = 1$ to $N$ **do**
2:    **for** each $(c, I_{\text{ref}})$ in dataset **do**
3:       Randomly select a timestep $t \sim \text{Uniform}(1, T_{\max})$
4:       Get the latent representations: $z_t \leftarrow \epsilon_\theta(c, \epsilon, t)$,
5:       Add noise to the $I_{\text{ref}}$ at step $t$:
6:       $z_{\text{ref},t} = \sqrt{\alpha_t} I_{\text{ref}} + \sqrt{1 - \alpha_t}\epsilon$
7:       **for** layer $l \in L_{\text{sel}}$ **do**
8:          Extract $\{Q_l^t, K_l^t, V_l^t\}$ from $z_t$ using $\theta$
9:          Extract $\{Q_l^{\text{ref},t}, K_l^{\text{ref},t}, V_l^{ref,t}\}$ from $z_{\text{ref},t}$ using $\theta_{ori}$
10:      **end for**
11:      Compute HAD loss by leveraging (4)
12:      Compute content-preserving loss by leveraging (5)
13:      Compute generalization-retain loss by leveraging (6)
14:      Compute total loss by leveraging (7)
15:      Update model parameters: $\theta \leftarrow \theta - \eta \nabla_\theta \mathcal{L}_{\text{total}}$
16:    **end for**
17: **end for**
18: **return** $\theta$

---

## A.2 IMPLEMENTATION DETAILS.

All experiments are conducted using the Stable Diffusion v1.5 model CompVis (2022) as the backbone. Stable Diffusion V2-1 is also avaliable. In our approach, only the self-attention layers of the UNet in the stable diffusion model are fine-tuned. Empirically, we use the first 10 self-attention layers of the UNet to compute the $\mathcal{L}_{HAD}$ and $\mathcal{L}_{content}$. The model is then trained for 50 epochs using a batch size of 1 and a learning rate of $1 \times 10^{-5}$. We set the loss weights to $\lambda_1 = 0.5$ and $\lambda_2 = 0.000325$.

## A.3 USER STUDY

Following ESDGandikota et al. (2023), we conduct a user study to assess how well different methods erase or preserve artistic styles for generalization. Each participant views five real artworks in a consistent style—either the target style or a generalization style—along with a sixth image, randomly selected as either: *1)* an image with the target style removed by SUDM or a baseline, or *2)* an image retaining a different style after target-style unlearning. A total of 2,925 responses are collected from 13 participants, each rating the similarity of the sixth image to the reference set on a scale from 0 to 5 (most similar). Figure 5 reports the average user ratings. SUDM obtains the lowest score on the target-style set, indicating strong erasure, and a high score on the generalization-style set, suggesting effective preservation. These results highlight SUDM's advantage in style-specific unlearning.

## A.4 SENSITIVITY ANALYSIS

To better understand the role of different components in our objective, we conduct a sensitivity analysis on the loss weights $\lambda_1$ (content preservation) and $\lambda_2$ (generalization preservation). The results are as follows:

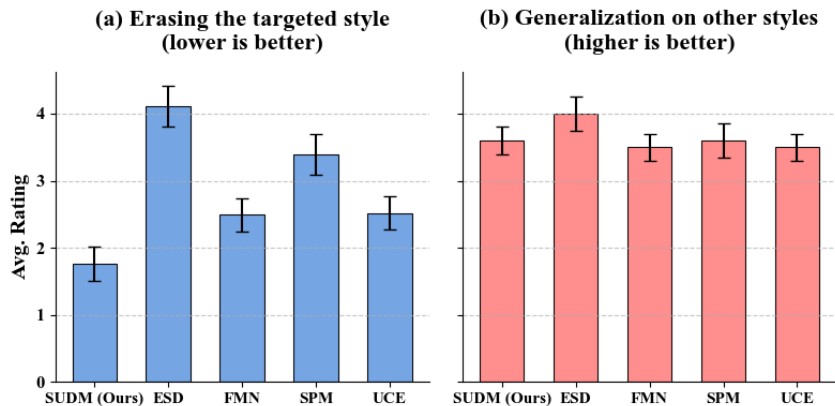

Figure 5: User study compares SUDM and baselines on erasing target styles and preserving others.

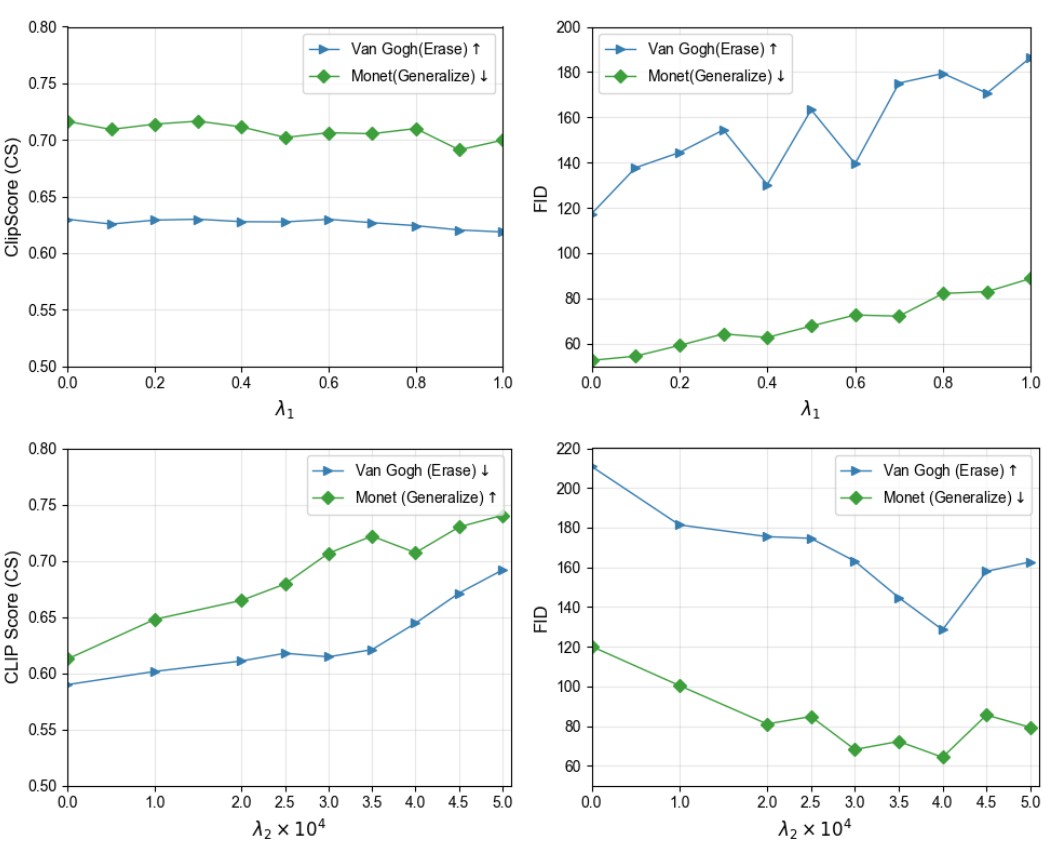

Figure 6: Sensitivity analysis of loss weights $\lambda_1$ (content preservation) and $\lambda_2$ (generalization preservation) on Van Gogh erasure and Monet generalization. Top-left: Effect of $\lambda_1$ on CS; Top-right: Effect of $\lambda_1$ on FID. Bottom-left: Effect of $\lambda_2$ on CS; Bottom-right: Effect of $\lambda_2$ on FID.

## A.5 THE INFLUENCE OF SELF-ATTENTION LAYERS SELECTION

To investigate the impact of layer selection on style forgetting, we conducted an ablation study by progressively incorporating self-attention layers into the optimization. Specifically, we trained models with cumulative ranges of layers, starting from 0–1 and gradually extending up to 0–16. As shown in Fig.7, the erasure becomes increasingly effective as more layers are involved, confirming that style information is distributed throughout the hierarchy rather than confined to a single depth.

However, we observe that applying optimization to all 16 layers, while yielding the strongest forgetting signal, also introduces overfitting, manifested as reduced diversity and fidelity in the generated images. Based on these findings, we adopt the first ten layers in our final configuration, which provides a favorable trade-off between effective style erasure and preservation of overall generative quality.

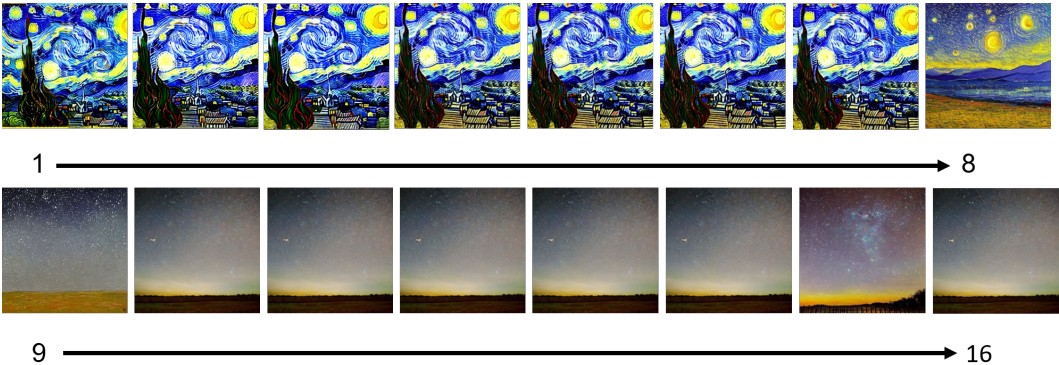

1 ⟶ 8

9 ⟶ 16

Figure 7: Ablation on the selection of self-attention layers for HAD. We progressively expand the included range from 0–1 up to 0–16. The erasure improves as more layers are involved, but full-layer optimization (0–16) leads to overfitting and degraded image fidelity. We therefore adopt the first ten layers as a practical trade-off.

## A.6 SELF-ATTENTION AND CROSS-ATTENTION IN STABLE DIFFUSION

In latent diffusion architectures (e.g., Stable Diffusion), cross-attention and self-attention serve different but interdependent functions. As can be see in Fig,8, Cross-attention conditions the visual latents on the textual embedding, injecting semantic guidance and any stylistic descriptors into the generative process. Self-attention, by contrast, operates exclusively within the visual latent space, modeling long-range spatial dependencies and enforcing coherent propagation of locally-introduced attributes across the image. Consequently, although stylistic cues can be introduced via the prompt through cross-attention, their spatial manifestation and global consistency are mediated by the self-attention pathway. Building on this distinction, we modify the self-attention mechanism to erase the style-related features while retaining semantic guidance from cross-attention. This intervention enables targeted style erasure, allowing the model to generate content faithful to the prompt without reproducing its stylistic attributes. We empirically demonstrate that this intervention effectively removes target styles while preserving semantic fidelity and overall image quality.

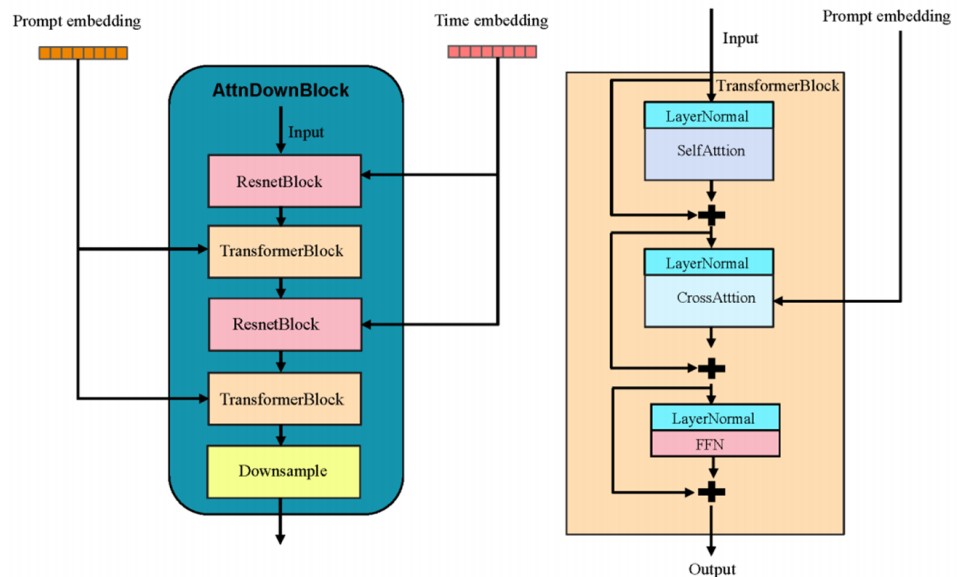

Figure 8: The self-attention and cross-attention in Stable Diffusion.

### A.7 ETHICS STATEMENT

This work adheres to the ICLR Code of Ethics. In this study, no human subjects or animal experimentation was involved. All datasets used were sourced in compliance with relevant usage guidelines, ensuring no violation of privacy. We have taken care to avoid any biases or discriminatory outcomes in our research process. No personally identifiable information was used, and no experiments were conducted that could raise privacy or security concerns. We are committed to maintaining transparency and integrity throughout the research process.

### A.8 REPRODUCIBILITY STATEMENT

We have made every effort to ensure that the results presented in this paper are reproducible. All code and datasets have been made publicly available in an anonymous repository to facilitate replication and verification. The experimental setup, including training steps, model configurations, and hardware details, is described in detail in the paper. We have also provided a full description of SUDM, to assist others in reproducing our experiments.

### A.9 THE USE OF LARGE LANGUANGE MODELS (LLM)

In preparing this manuscript, LLMs were used solely for two auxiliary tasks,

1. **Language Polishing**: [ GPT-4] was employed to refine sentence structure, enhance expression clarity, and ensure writing style consistency. All LLM-generated revisions were manually checked and adjusted to align with academic norms and the original research intent.

2. **Literature Summary Assistance**: The LLM assisted in synthesizing key findings from peer-reviewed literature (independently retrieved and screened by the authors). These summaries served only as preliminary references; the authors further cross-verified content against original sources and conducted critical analysis to finalize the literature review.

