# OpenReview forum: "Style Unlearning in Diffusion Models"
_ICLR.cc/2026/Conference — ICLR 2026 Conference Withdrawn Submission_

### Official Review · Reviewer_edyT · 2025-10-29

**Soundness:** 3
**Presentation:** 3
**Contribution:** 2
**Rating:** 4
**Confidence:** 4

**Summary:**

This paper proposes SUDM (Style Unlearning in Diffusion Models), an attention-based framework that removes specific artistic styles from diffusion models while preserving semantic and visual quality. The method leverages the observation that self-attention keys/values encode style and queries encode content, introducing a Hybrid Attention Distillation loss combining style erasure, content preservation, and regularization.

**Strengths:**

1. The proposed approach reframes style removal as an attention-space alignment problem.
2. The proposed Hybrid-Attention Distillation (HAD) exploits the empirical observation that self-attention keys/values encode style while queries preserve semantic content

**Weaknesses:**

1. Each training pair requires a style-neutral reference image that shares content with the stylized prompt. This paper uses text-to-image generation with negative prompts (“no Van Gogh”), which may still contain mild stylistic residues.
2. Limited scope of evaluation. More strong unlearning baselines such as AdvUnlearn [1] should be considered. Furthermore, robustness evaluation through adversarial attack such as UnlearnDiffAtk [2] should be involved.
3. CLIP Score and FID indirectly reflect semantic retention but conflate content and style. Including LPIPS or DINO-based content similarity metrics would provide clearer insight into semantic preservation.

[1] Defensive Unlearning with Adversarial Training for Robust Concept Erasure in Diffusion Models, NeurIPS 2024

[2] To Generate or Not? Safety-Driven Unlearned Diffusion Models Are Still Easy To Generate Unsafe Images … For Now, ECCV 2024

**Questions:**

1. Have you tried to visualize attention heatmaps before and after unlearning to confirm that K,V features indeed correspond to stylistic texture regions?
2. How does SUDM behave when unlearning more than two styles (e.g., five)? Does the parameter regularization term remain sufficient?

---

### Official Review · Reviewer_nBQk · 2025-10-30

**Soundness:** 3
**Presentation:** 3
**Contribution:** 2
**Rating:** 6
**Confidence:** 4

**Summary:**

This paper proposes a training-based method for erasing style concept using diffusion models. The contribution includes the model designs/losses in the attention layers (e.g., Hybrid-Attention Distillation, content and generalization preservation), the proposed dataset constrution and evalutation results and analysis.

**Strengths:**

1. the paper is logically fluent and the writing is good.
2. The experimental design is thorough.
3. The experiment results validates the techinical contribution.
4. The framework supports simultaneous unlearning of multiple styles (e.g., Van Gogh and Monet) while maintaining the model's performance on unrelated styles and content.

**Weaknesses:**

1. Evaluation metric: clip score could contain both style and content information so not necessarily lower the better.
2. The design is based on assumption that  "Q encapsulates the content within an image, while K and V represent the style information" (L209), any experiments to validate this assumption?
3. The dataset construction relies on deriving a base prompt and using a negative prompt to generate the style-neutral reference image. How can you ensure precise pixel-level or contour alignment between the stylized image and the generated style-neutral reference image for paired distillation?
4. How does SUDM prevent the generation of the target style when a user employs indirect references (e.g., "Post-Impressionist era" or "Expressionism" to refer to Van Gogh)?
5. The final configuration (A.2) uses the first 10 self-attention layers for optimization , based on a trade-off analysis. Is there a theoretical or architectural justification for why style entanglement is stronger/weaker in the later, deeper self-attention layers of the U-Net?

**Questions:**

add questions in weakness section.

---

### Official Review · Reviewer_v54q · 2025-11-04

**Soundness:** 2
**Presentation:** 3
**Contribution:** 2
**Rating:** 4
**Confidence:** 4

**Summary:**

This work proposes SUDM (Style Unlearning in Diffusion Models), a new framework designed to remove stylistic representations from pretrained diffusion models (e.g., Stable Diffusion) while preserving semantic and generative quality. The key innovation lies in Hybrid-Attention Distillation (HAD), which aligns self-attention (encoding style) with cross-attention (style-neutral) to erase style-specific features. To prevent semantic and generalization degradation, the method introduces query consistency and parameter retain losses.
Experiments on multiple artist styles (Van Gogh, Monet, Picasso, Rembrandt) show that SUDM outperforms prior unlearning approaches (ESD, FMN, SPM, UCE) in both visual fidelity and style erasure, with quantitative metrics (CLIPScore, FID) and user studies.

**Strengths:**

- The proposed method leverages cross-attention as style-neutral supervision for self-attention unlearning. The method is simple to reproduce an verify.
- The paper is well-organized  and clearly written. It includes clear mathematical definitions for all loss components.
- The study is relatively comprehensive, including single-style and multi-style unlearning, ablation tests, sensitivity analyses, and user studies.

**Weaknesses:**

- The method assumes a clear separation of style and content across Q (content) vs. K/V (style) in self-attention. While supported by prior works, this assumption may not hold in latest new diffusion models, such as DiT, SDXL, PixArt, DeepFloyd etc. More experiments are needed to show the generalization ability of this method.
- Experiments are restricted to four canonical Western painters with highly distinctive styles. It is unclear how the approach performs on modern, hybrid, or less distinctive styles, or on non-artistic style domains (e.g., photography filters, cartoon styles).
- The training process assumes access to a reference image with identical semantics but a neutral style. This requirement may limit practical applicability, as such references are rarely available for arbitrary prompts or artists.

**Questions:**

For other new diffusion models, especially recent transformer based ones, such as DiT, is this proposed method still working?

---

### Official Review · Reviewer_SSYE · 2025-11-07

**Soundness:** 1
**Presentation:** 2
**Contribution:** 1
**Rating:** 2
**Confidence:** 4

**Summary:**

Machine unlearning is crucial for addressing intellectual property and ethical concerns arising from unauthorized style replication. However, most existing unlearning methods struggle to completely remove target styles while preserving generation quality.
This paper introduces a framework based on **hybrid-attention distillation**, where cross-attention provides style-agnostic supervision to self-attention for targeted style erasure. To ensure content preservation and robust generalization, the method incorporates **query consistency** and **parameter consistency** losses into the training objective.
Experiments conducted across **four artists** and over **400 generated images** using **20 test prompts** demonstrate the method’s effectiveness in removing artistic styles while maintaining high visual fidelity. The experimental results do not fully support the major claims in the paper, raising concerns about the effectiveness of the proposed method.

**Strengths:**

The problem is important, and the proposed approach is interesting, even though there are some major technical flaws in its definition and evaluation. The authors conduct **extensive experiments** to evaluate the effectiveness of their proposed method, providing valuable empirical insights into the style unlearning problem within diffusion-based generative models.

**Weaknesses:**

1. Lines 79–82:
   The description “...hybrid-attention distillation leverages the key (K) and value (V) from the reference image as style-neutral anchors, and the query (Q) from the generated image to encode content information. Then, a cross-attention mechanism between Q and the style-neutral K, V guides the self-attention in the generated image to unlearn the style.” raises several concerns.
   - The proposed method requires both style-neutral anchors and content information from the generated image, which could be conflicting.
   - It is unclear at this point whether the query is derived from **intermediate time steps** or the **fully generated image**.
   - Finding truly style-neutral anchors is also challenging, as multiple artistic styles can share similar anchor representations.

2. Lines 91–92:
   The statement “To the best of our knowledge, it is the first unlearning technique tailored for style removal from diffusion models.” is **factually incorrect**. Several prior works already address unlearning concepts (including styles) in diffusion models, such as **Gandikota et al., ICCV 2023; Kumari et al., ICCV 2023; Wu et al., 2024 (EraseDiff)**, among others. The authors should revise this claim and properly acknowledge related works.

3. Typo in Line 134:
   Remove the extra space in “layers ,” to read as “layers,”.

4. Lines 223–225:
   The claim that “this design ensures that the model focuses on erasing style-related features, while preserving semantic content encoded primarily in the query representations” is not well justified.
   - What variables are optimized in Equation (4)?
   - The function $\mathcal{L}_{HAD}$ and its variables should be explicitly defined.
   - Without further analysis, it cannot be concluded that content and style are separable in the query representations.
   Additional empirical or theoretical evidence is needed to support this assumption.

5. Equation (7):
   The total loss and its components are presented without explicitly defining the variables involved. Please clarify what parameters or embeddings each loss term depends on (Although one can infer from the context, it is important to define this properly for mathematical clarity).

6. Inconsistencies and unclear evaluation in Table 2:
   - Typo: ESD performing better than UCE in CS for Picasso seems inconsistent with the qualitative discussion.
   - The claim that the method forgets Van Gogh while preserving Picasso, Monet, and Rembrandt is not fully supported. The FID scores indicate that **Picasso style** is also degraded, with preservation performance lower than UCE despite comparable CLIP scores. Similar inconsistencies appear for Monet and Rembrandt.

7. Preservation quality issues in Table 3:
   The pattern repeats across Tables 1–4. While the proposed method achieves reasonable style erasure (CLIP and FID scores), **preservation quality is significantly worse**, with FID values more than twice for some of the comparable baselines. This suggests that the method sacrifices generation fidelity for unlearning performance.

8. Lines 349–350:
   Instead of referring generically to “Tables 1–4,” the authors should specify the unique findings from each table and summarize the key insights drawn from those results.

9. Table 4 results show that the proposed method struggles with **multi-style forgetting**, where the baseline UCE consistently outperforms the proposed model. This contradicts claims of robust generalization.

**Questions:**

Please see the weakness section above.

---

### Note · Authors · 2025-11-14

I have read and agree with the venue's withdrawal policy on behalf of myself and my co-authors.